# The Met Dataset:
# Instance-level Recognition for Artworks

**Nikolaos-Antonios Ypsilantis**
VRG, Faculty of Electrical Engineering
Czech Technical University in Prague

**Noa Garcia**
Institute for Datability Science
Osaka University

**Guangxing Han**
DVMM Lab
Columbia University

**Sarah Ibrahimi**
Multimedia Analytics Lab
University of Amsterdam

**Nanne van Noord**
Multimedia Analytics Lab
University of Amsterdam

**Giorgos Tolias**
VRG, Faculty of Electrical Engineering
Czech Technical University in Prague

## Abstract

This work introduces a dataset for large-scale instance-level recognition in the domain of artworks. The proposed benchmark exhibits a number of different challenges such as large inter-class similarity, long tail distribution, and many classes. We rely on the open access collection of The Met museum to form a large training set of about 224k classes, where each class corresponds to a museum exhibit with photos taken under studio conditions. Testing is primarily performed on photos taken by museum guests depicting exhibits, which introduces a distribution shift between training and testing. Testing is additionally performed on a set of images not related to Met exhibits making the task resemble an out-of-distribution detection problem. The proposed benchmark follows the paradigm of other recent datasets for instance-level recognition on different domains to encourage research on domain independent approaches. A number of suitable approaches are evaluated to offer a testbed for future comparisons. Self-supervised and supervised contrastive learning are effectively combined to train the backbone which is used for non-parametric classification that is shown as a promising direction. Dataset webpage: `http://cmp.felk.cvut.cz/met/`.

## 1 Introduction

Classification of objects can be done with categories defined at different levels of granularity. For instance, a particular piece of art is classified as the "Blue Poles" by Jackson Pollock, as painting, or artwork, from the point of view of instance-level recognition [8], fine-grained recognition [24], or generic category-level recognition [32], respectively. Instance-level recognition (ILR) is applied to a variety of domains such as products, landmarks, urban locations, and artworks. Representative examples of real world applications are place recognition [1, 22], landmark recognition and retrieval [39], image-based localization [33, 3], street-to-shop product matching [2, 17, 26], and artwork recognition [11]. There are several factors that make ILR a challenging task. It is typically required to deal with a large category set, whose size reaches the order of $10^6$, with many classes represented by only a few or a single example, while the small between class variability further increases the hardness. Due to these difficulties the choice is often made to handle instance-level classification as an instance-level retrieval task [37]. Particular applications, *e.g.* in the product or art

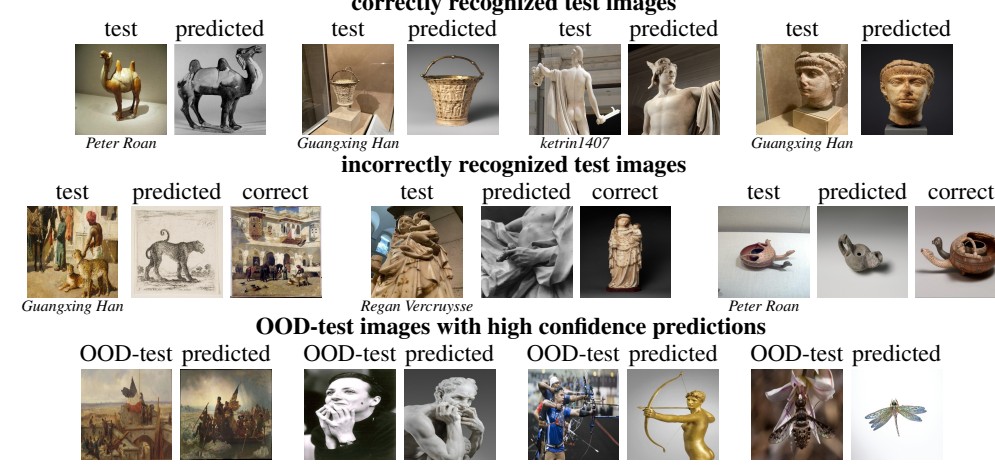

Figure 1: Challenging examples from the Met dataset for the top performing approach. Test images are shown next to their nearest neighbor from the Met exhibits that generated the prediction of the corresponding class. Top row: correct predictions. Middle row: incorrect predictions; an image of the ground truth class is also shown. Bottom row: high confidence predictions for OOD-test images; the goal is to obtain low confidence for these.

domain require dynamic updates of the category set; images from new categories are continuously added. Therefore, ILR is a form of open set recognition [16].

Despite the many real-world applications and challenging aspects of the task, ILR has attracted less attention than category-level recognition (CLR) tasks, which are accompanied by large and popular benchmarks, such as ImageNet [31], that serve as a testbed even for approaches applicable beyond classification tasks. A major cause for this is the lack of large-scale datasets. Creating datasets with accurate ground truth at large scale for ILR is a tedious process. As a consequence, many datasets include noise in their labels [8, 11, 39]. In this work, we fill this gap by introducing a dataset for instance-level classification in the artwork domain.

The art domain has attracted a lot of attention in computer vision research. A popular line of research focuses on a specific flavor of classification, namely attribute prediction [21, 27, 28, 36, 40]. In this case, attributes correspond to various kinds of metadata for a piece of art, such as style, genre, period, artist and more. The metadata for attribute prediction is obtained from museums and archives that make this information freely available. This makes the dataset creation process convenient, but the resulting datasets are often highly noisy due to the sparseness of this information [27, 36]. Another known task is domain generalization or adaptation where object recognition or detection models are trained on natural images and their generalization is tested on artworks [10]. A very challenging task is motif discovery [34, 35] which is intended as a tool for art historians, and aims to find shared motifs between artworks. In this work we focus on ILR for artworks which combines the aforementioned challenges of ILR, is related to applications with positive impact, such as educational applications, and has not yet attracted attention in the research community.

We introduce a new large-scale dataset (see Figure 1 for examples) for instance-level classification by relying on the open access collection from the Metropolitan Museum of Art (The Met) in New York. The training set consists of about 400k images from more than 224k classes, with artworks of world-level geographic coverage and chronological periods dating back to the Paleolithic period. Each museum exhibit corresponds to a unique artwork, and defines its own class. The training set exhibits a long-tail distribution with more than half of the classes represented by a single image, making it a special case of few-shot learning. We have established ground-truth for more than $1,100$ images from museum visitors, which form the query set. Note that there is a distribution shift between this query set and the training images which are created in studio-like conditions. We additionally include a large set of distractor images not related to The Met, which form an Out-Of-Distribution (OOD) [25, 30] query set. The dataset follows the paradigm and evaluation protocol of the recent Google Landmarks Dataset (GLD) [39] to encourage universal ILR approaches that are applicable in a wider range of domains. Nevertheless, in contrast to GLD, the established ground-

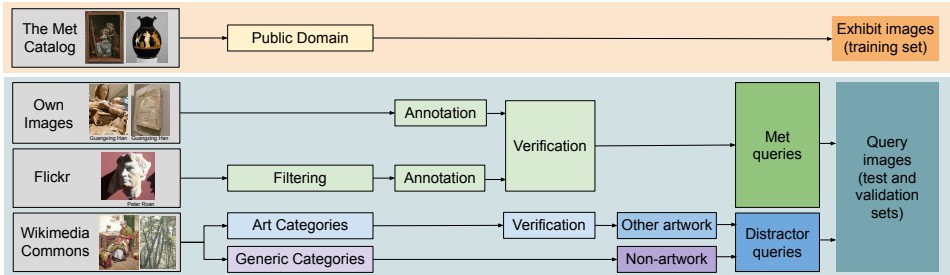

Figure 2: The Met dataset collection and annotation process.

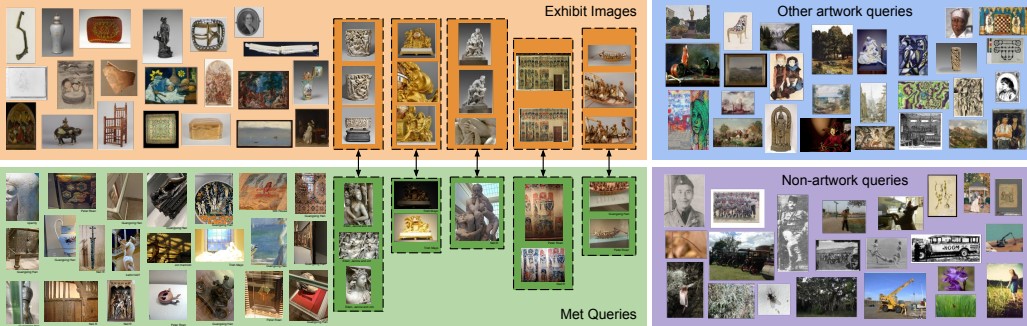

Figure 3: Samples from the Met dataset of exhibit and query (Met and distractor) images, demonstrating the diversity in viewpoint, lighting, and subject matter of the images. Exhibit images and queries from the same Met class are indicated by dashed lines.

truth does not include noise. To our knowledge this the only ILR dataset at this scale, that includes no noise in the ground-truth and is fully publicly available.

The introduced dataset is accompanied by performance evaluation of relevant approaches. We show that non parametric classifiers perform much better than parametric ones. Improving the visual representation becomes essential with the use of non-parametric classifiers. To this end, we show that the recent self-supervised learning methods that rely only on image augmentations are beneficial, but the available ILR labels should not be discarded. A combined self-supervised and supervised contrastive learning approach is the top performer in our benchmark indicating promising future directions.

## 2 The Met dataset

The *Met dataset* for ILR contains two types of images, namely *exhibit images* and *query images*. Exhibit images are photographs of artworks in The Met collection taken by The Met organization under studio conditions, capturing multiple views of objects featured in the exhibits. These images form the training set for classification and are interchangeably called exhibit or training images in the following. We collect about 397k exhibit images corresponding to about 224k unique exhibits, *i.e.* classes, also called *Met classes*.

Query images are images that need to be labeled by the recognition system, essentially forming the evaluation set. They are collected from multiple online sources for which ground-truth is established by labeling them according to the Met classes. The Met dataset contains about 20k query images, that are divided into the following three types: 1) *Met queries*, which are images taken at The Met museum by visitors and labeled with the exhibit depicted, 2) *other-artwork queries*, which are images of artworks from collections that do not belong to The Met, and 3) *non-artwork queries*, which are images that do not depict artworks. The last two types of queries are referred to as *distractor queries* and are labeled as "distractor" class which denotes out-of-distribution queries.

**Dataset collection.** The dataset collection and annotation process is described in the following and summarized in Figure 2, while sample images from the dataset are shown in Figure 3.

| Split | Type | # Images | | | # Classes |
| --- | --- | --- | --- | --- | --- |
| | | Met | other-art | non-art | |
| Train | Exhibit | 397, 121 | - | - | 224, 408 |
| Val | Query | 129 | 1, 168 | 868 | 111 + 1 |
| Test | Query | 1, 003 | 10, 352 | 7, 964 | 734 + 1 |

Table 1: Number of images and classes in the Met dataset per split. Met exhibits images are from the museum's open collection, while Met query images are from museum visitors. Query images contain distractor images too (denoted by the +1 class) while the rest of val/test classes are subset of the train classes.

*Image sources:* Exhibit images are obtained from The Met collection.[1] Only exhibits labeled as open access are considered. A maximum of 10 images per exhibit is included in the dataset, images with very skewed aspect ratios are excluded, and image deduplication is performed. Query images are collected from different sources according to the type of query. Met queries are taken on site by museum visitors. Part of them are collected by our team, and the rest are Creative Commons (CC) images crawled from Flickr. We use Flickr groups[2] related to The Met to collect candidate images. Distractor queries are downloaded from Wikimedia Commons[3] by crawling public domain images according to the Wikimedia assigned categories. Generic categories, such as people, nature, or music, are used for non-artwork queries, and art-related categories, *e.g*. art, sculptures, painting, architecture, for other-artwork queries.

*Annotation:* We label query images with their corresponding Met class, if any. Met queries taken by our team are annotated based on exhibit information, whereas Met queries downloaded from Flickr are annotated in three phases, namely filtering, annotation, and verification. In the filtering phase, invalid images are discarded, *i.e*. images containing visitor faces, images not depicting exhibits, or images with more than one exhibit. In the annotation phase, queries are labeled with the corresponding Met class. To ease the task, the title and description fields on Flickr are used for text-based search in the list of titles from The Met exhibits included in the corresponding metadata. Queries whose depicted Met exhibit is not in the public domain are discarded. Finally, in the verification phase, two different annotators verify the correctness of the labeling per query. We additionally verify that distractor queries, especially other-artwork queries, are true distractors and do not belong to The Met collection. This is done in a semi-automatic manner supported by (i) text-based filtering of the Wikimedia image titles and (ii) visual search using a pre-trained deep network. Top matches are manually inspected and images corresponding to Met exhibits are removed.

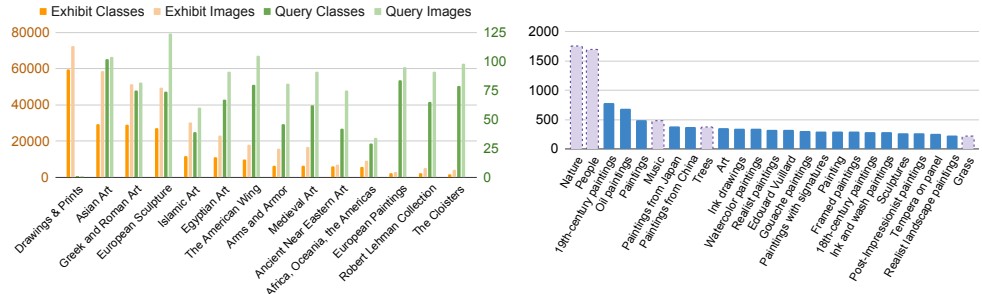

Figure 4: Left: Number of images and classes by department. Met queries are assigned to the department of their ground-truth class. Some departments that do not contain queries but contain exhibit images are not shown. Right: Number of distractor images by Wikimedia category. Top categories shown: art-related categories in solid blue and generic categories in dash purple.

**Benchmark and evaluation protocol.** The structure and evaluation protocol for the Met dataset follows that of the Google Landmarks Dataset (GLD) [39]. All Met exhibit images form the training

[1]https://www.metmuseum.org/

[2]https://www.flickr.com/groups/metmuseum/, https://www.flickr.com/groups/themet/, https://www.flickr.com/groups/mma_aaoa/

[3]https://commons.wikimedia.org/wiki/Main_Page

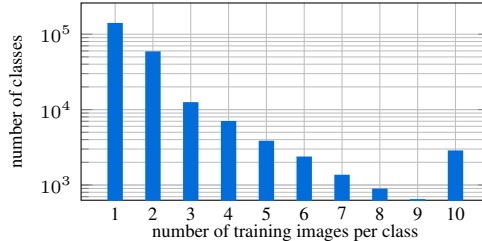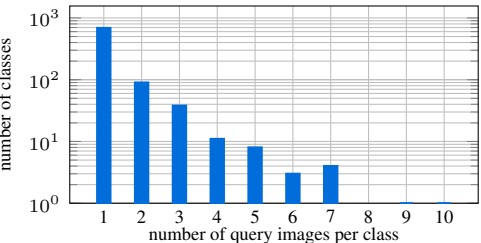

Figure 5: Left: number of Met classes versus number of training images per class. Right: number of Met classes versus number of query images per class.

set, while the query images are split into test and validation sets. The test set is composed of roughly 90% of the query images, and the rest is used to form the validation set. To ensure no leakage between the validation and test split, all Met queries are first grouped by user and then assigned to a split. Additionally, we enforce that there is no class overlap between the splits. As a result, 25 (14) users appear only in the test (validation) split, respectively. Image and class statistics for the train, val, and test parts are summarized in Table 1. The intended use of the validation split is for hyper-parameter tuning. All images are resized to have maximum resolution $500 \times 500$.

For evaluation we measure the classification performance with two standard ILR metrics, namely average classification accuracy (ACC), and Global Average Precision (GAP). The average classification accuracy is measured only on the Met queries, whereas the GAP, also known as Micro Average Precision ($\mu$AP), is measured on all queries taking into account both the predicted label and the prediction confidence. All queries are ranked according to the confidence of the prediction in descending order, and then average precision is estimated on this ranked list; predicted labels and ground-truth labels are used to infer correctness of the prediction, while distractors are always considered to have incorrect predictions. GAP is given by $\frac{1}{M} \sum_{i=1}^{T} p(i)r(i)$, where $p(i)$ is the precision at position $i$, $r(i)$ is a binary indicator function denoting the correctness of prediction at position $i$, $M$ is the number of the Met queries, and $T$ is the total number of queries. The GAP score is equal to the area-under-the-curve of the precision-recall curve whilst jointly taking all queries into account. We measure this for the Met queries only, denoted by GAP$^-$, and for all queries, denoted by GAP. In contrast to accuracy, this metric reflects the quality of the prediction confidence as a way to detect out-of-distribution (distractor) queries and incorrectly classified queries. It allows for inclusion of distractor queries in the evaluation without the need for distractors in the learning; the classifier never predicts "out-of-Met" (distractor) class. Optimal GAP requires, other than correct predictions for all Met queries, that all distractor queries get smaller prediction confidence than all the Met queries.

**Dataset statistics.** The Met dataset contains artworks spanning from as far back as $240,000$ BC to the current day. Figure 4 (left) shows the distribution of classes and images according to The Met department. Whereas there is an imbalance for exhibits across The Met departments, queries are collected to be evenly distributed to the best of our capabilities. In this way, we aim to ensure models are not biased towards a specific type of art, i.e., developing models that only produce good results for, e.g., European paintings, will not necessarily ensure good results on the overall benchmark. Finally, Figure 4 (right) shows the number of distractor query images by Wikimedia Commons categories.

The class frequency for exhibit images ranges from 1 to 10, with $60.8\%$ and $1.2\%$ classes containing a single and 10 images, respectively (see Figure 5 left). Met queries are obtained from 39 visitors in total, while the maximum number of query images per class is, coincidentally, also 10. In total, $81.5\%$ of the Met query images are the sole Met queries that depict a particular Met class (see Figure 5 right).

**Comparison to other datasets.** We compare the Met dataset with existing datasets that are relevant in terms of domain or task.

*Artwork datasets:* Table 2 summarizes datasets in the artwork domain for various tasks. Most of the artwork datasets [21, 27, 28, 36, 40] focus on attribute prediction (AP), containing multiple types of annotations, such as author, material, or year of creation, usually obtained directly from the museum collections. Other datasets [5, 10, 40, 42] are focused on CLR, aiming to recognize object

| Art datasets | Year | Domain | # Images | # Classes | Type of annotations | Task | Image source |
|---|---|---|---|---|---|---|---|
| PrintArt [5] | 2012 | Prints | 988 | 75 | Art theme | CLR | Artstor |
| VGG Paintings [10] | 2014 | Paintings | 8,629 | 10 | Object category | CLR | Art UK |
| WikiPaintings [21] | 2014 | Paintings | 85,000 | 25 | Style | AP | WikiArt |
| Rijksmuseum [28] | 2014 | Artwork | 112,039 | †6,629 | Art attributes | AP | Rijksmuseum |
| BAM [40] | 2017 | Digital art | 65M | †9 | Media, content, emotion | AP, CLR | Enhance |
| Art500k [27] | 2017 | Artwork | 554,198 | †1,000 | Art attributes | AP | Various |
| SemArt [14] | 2018 | Paintings | 21,383 | 21,383 | Art attributes, descriptions | Text-image | Web Gallery of Art |
| OmniArt [36] | 2018 | Artwork | 1,348,017 | †100,433 | Art attributes | AP | Various |
| Open MIC [23] | 2018 | Artwork | 16,156 | 866 | Instance | ILR (DA) | Authors |
| iMET [42] | 2019 | Artwork | 155,531 | 1,103 | Concepts | CLR | The Met |
| NoisyArt [11] | 2019 | Artwork | 89,095 | 3,120 | Instance (noisy) | ILR | Various |
| **The Met (Ours)** | 2021 | Artwork | 418,605 | 224,408 | Instance | ILR | Various |

Table 2: Comparison to art datasets. † For datasets with multiple annotations, the task with the largest number of classes is reported.

| ILR datasets | Year | Domain | # Images | # Classes | Type of annotations | Image source |
|---|---|---|---|---|---|---|
| Street2Shop [17] | 2015 | Clothes | 425,040 | 204,795 | Category, instance | Various |
| DeepFashion [26] | 2016 | Clothes | 800,000 | 33,881 | Attributes, landmarks, instance | Various |
| GLD v2 [39] | 2019 | Landmarks | 4.98M | 200,000 | Instance (noisy) | Wikimedia |
| AliProducts [8] | 2020 | Products | 3M | 50,030 | Instance (noisy) | Alibaba |
| Products-10K [2] | 2020 | Products | 150,000 | 10,000 | Category, instance | JD.com |
| **The Met (Ours)** | 2021 | Artwork | 418,605 | 224,408 | Instance | Various |

Table 3: Comparison to instance-level recognition datasets.

categories, such as animals and vehicles, in paintings. From the artwork datasets, Open MIC [23] and NoisyArt [11] are the only ones with instance-level labels. Compared to the Met dataset, the Open MIC is smaller, with significantly less classes and mostly focuses on domain adaptation (DA) tasks. NoisyArt has a similar focus to ours, but is significantly smaller, and has noisy labels.

*ILR datasets:* In Table 3 we compare the Met dataset with existing ILR datasets in multiple domains. ILR is widely studied for clothing [17, 26], landmarks [39], and products [2, 8]. The Met dataset resembles ILR datasets in those domains in that the training and query images are from different scenarios. For example, in Street2Shop [17] and DeepFashion [26] queries are taken by customers in real-life environments, whereas training images are studio shots. Getting annotations for ILR, however, is not easy, and some datasets contain a significant number of noisy annotations from crawling from the web without verification [8, 11, 39]. In that sense, the Met is the largest ILR dataset in terms of number of classes, which have been manually verified. Overall, the Met dataset proposes a large-scale challenge in a new domain, encouraging future research on generic ILR approaches that are applicable in a universal way to multiple domains.

## 3 Baseline approaches

This section presents the approaches considered as baselines, *i.e.* existing methods that are applicable to this dataset, in the experimental evaluation.

**Representation.** Consider an embedding function $f_\theta : \mathcal{X} \to \mathbb{R}^d$ that takes an input image $x \in \mathcal{X}$ and maps it to a vector $f_\theta(x) \in \mathbb{R}^d$, equivalently denoted by $f(x)$. Function $f(\cdot)$ comprises a fully convolutional network (the backbone network), a global pooling operation that maps a 3D tensor to a vector, vector $\ell_2$ normalization, and an optional fully-connected layer (also seen as $1 \times 1$ convolution), and a final vector $\ell_2$ normalization. The backbone is parametrized by the parameter set $\theta$. ResNet18 (R18) and ResNet50 (R50) [18] are the backbones used in this work, while global pooling is performed by Generalized-Mean (GeM) pooling [29], shown to be effective for representation in instance-level tasks [4].

Representation of image $x$, denoted by vector embedding $\mathbf{v}(x) \in \mathbb{R}^d$, is a result of aggregation of multi-resolution embeddings given by

$$\mathbf{v}(x) = \frac{\sum_{r \in R} f(x_r)}{|| \sum_{r \in R} f(x_r)||}, \tag{1}$$

where $x_r$ denotes image $x$ down-sampled by relative factor $r$. We set $R = \{1, 2^{-0.5}, 2^{-1}\}$ and $R = \{1\}$ in the *multi-scale* (MS) and *single-scale* (SS) case, respectively. Following the standard practice in instance-level search, the image representation space is whitened with PCA whitening

(PCAw) [20] learned on the representation vectors of all Met training images. Optionally, dimensionality reduction is performed by keeping the dimensions corresponding to the top components. PCAw is always performed in the rest of the paper, unless stated otherwise; for simplicity we reuse notation $\mathbf{v}(x)$ for the whitened image embeddings. Given a trained backbone (fixed $\theta$), the image representation is consequently used in combination with a k-Nearest-Neighbor (kNN) classifier.

**kNN classifier.** The label of image $x$ is denoted by $y(x)$ and $q$ is a query image. The similarity between query and a training image is given by $\mathbf{v}(x)^\top \mathbf{v}(q)$, coinciding with the cosine similarity. The confidence of class $c$ for query $q$ is given by

$$s_c(q) = \max_{x \in \mathrm{NN}_k(q)} (\mathbf{v}(x)^\top \mathbf{v}(q)) \mathbb{1}_{y(x)=c}, \tag{2}$$

where $\mathrm{NN}_k(q)$ is the set of $k$ nearest-neighbors of $q$ in the $d$-dimensional representation space. The vector of class confidences is $\mathbf{s}(q) \in \mathbb{R}^N$ with elements $s_c(q), c \in [1, \dots, N]$, where $N$ is the number of training classes. Classes without any example in the top-$k$ neighbors have zero confidence. The predicted label $\hat{y}(q) = \arg\max_c s_c(q)$ is, according to (2), equivalent to the label of the closest training image. Despite label prediction requiring only $k = 1$, confidence estimation for more classes is essential for normalization and handling of OOD (distractor) queries. The normalized confidence is given by the soft-max of vector $\tau \mathbf{s}(q)$, where $\tau$ is the temperature. This is a non-parametric classifier that does not necessarily require training on The Met dataset; it only requires an existing backbone network. Hyper-parameters $k$ and $\tau$ are tuned with grid search according to GAP on the validation set.

**Training on the Met.** We use the Met training set and perform either training of a classifier for the Met classes or training of the backbone to obtain image embeddings for the kNN classifier. During all variants of training the backbone the optional FC layer is included in the architecture and initialized with the result of PCA whitening [29].

*Deep network (DNet) classifier with instance-level labels:* The backbone is trained jointly with a cosine similarity (linear) classifier [38], used previously for training with imbalanced datasets [19], combined with one of the two following losses. Cross-Entropy (CE) loss with soft-max, where the input to the soft-max is equal to the cosine similarity between the backbone output and the learnable class vectors (prototypes) multiplied by temperature $\gamma$. Alternatively, we use the ArcFace (AF) loss [12], which is also used in the work of Cao *et al.* [4] for instance-level recognition of landmarks. During inference two options are considered. First, use the whole deep network classifier and consider its $\arg\max$ and $\max$ as class prediction and confidence score, respectively. Second, discard the linear classifier and use the backbone $f_\theta(\cdot)$ to obtain the image representation $v(x)$ and make predictions with the kNN classifier.

*Simple-siamese (SimSiam) instance discrimination:* We apply the recent self-supervised approach by Chen and He [7] to train the backbone. Each training image is augmented twice resulting in a positive pair, while no negative pairs and no Met labels are used in this approach.

*Contrastive loss with synthetic/real positives and hard negatives:* The backbone is trained with contrastive loss [9], where each training image is used as an anchor to form one positive and one hard negative pair per epoch. A hard-negative pair is formed by randomly choosing an image among the 10 most similar images from a different class, as these are computed according to embeddings obtained with the current backbone before each epoch. Three different ways of forming the positive pair are tested. *Syn*: the positive is an augmented (synthesized) version of the anchor image. *Syn+Real*: the selected positive is another randomly chosen image of the same class as the anchor, or an augmented version of the anchor image. Synthetic positive or one of the real (all images in the class but the anchor) positives is chosen with equal probability which is equal to one over the number of images in the class. If the class has a single image, then augmentation is performed; note that many classes contain a single image. *Syn+Real-closest*: same as *Syn+Real* but the real positive counterpart is chosen to be the one with the most similar embedding to the anchor. This is used to avoid images that depict completely different views of the object and has previously been used in location estimation [1]. Synthetic or real positive is chosen with equal probability in this case.

**Pretrained models.** We consider networks pretrained on other tasks and use them to obtain the image embeddings for the kNN classifier. None of these variants includes the optional FC layer in the architecture.

| ID | Net | PCAw | MS | k | $\tau$ | GAP | GAP$^-$ | ACC |
|---|---|---|---|---|---|---|---|---|
| 1 | R18IN | | | 3 | 15 | 3.7 | 16.7 | 26.8 |
| 2 | R18IN | ✓ | | 7 | 100 | 10.9 | 28.0 | 33.7 |
| 3 | R18IN | | ✓ | 50 | 10 | 10.5 | 23.8 | 33.5 |
| 4 | R18IN | ✓ | ✓ | 3 | 50 | **15.9** | **37.5** | **42.3** |
| 5 | R18IN | ✓ | ✓ | 1 | - | 2.9 | 33.6 | 42.3 |
| 6 | R18IN† | ✓ | ✓ | 3 | 100 | 14.1 | 36.9 | 42.3 |

Table 4: Recognition performance for kNN classifier on representation obtained from ResNet18 pretrained on ImageNet. MS: multi-scale representation. †: tuning $k, \tau$ only with Met queries, and without distractor queries in the validation set.

| Net | GAP | GAP$^-$ | ACC |
|---|---|---|---|
| R18IN [18] | 15.9 (+0.0) | 37.5 (+0.0) | 42.3 (+0.0) |
| R18SFM [29] | 23.2 (+7.3) | 41.5 (+4.0) | 45.7 (+3.4) |
| R18SWSL [41] | **24.7 (+8.8)** | **47.0 (+9.5)** | **50.9 (+8.6)** |
| R50IN [18] | 22.2 (+0.0) | 41.8 (+0.0) | 46.4 (+0.0) |
| R50SFM [29] | 26.6 (+4.4) | 44.8 (+3.0) | 48.6 (+2.2) |
| R50SemArt (author) [13] | 1.8 (-20.4) | 12.2 (-29.6) | 18.0 (-28.4) |
| R50SemArt (type) [13] | 7.9 (-14.3) | 26.8 (-15.0) | 31.9 (-14.5) |
| R50SIN [15] | 15.5 (-6.7) | 36.4 (-5.4) | 41.7 (-4.7) |
| R50SwAV [6] | 22.8 (+0.6) | 45.0 (+3.2) | 49.6 (+3.2) |
| R50SWSL [41] | **30.4 (+8.2)** | **52.9 (+11.1)** | **56.3 (+9.9)** |

Table 5: Comparison of recognition performance for kNN classifier with representation from backbone networks pretrained for different tasks. Relative improvements compared to the corresponding network trained on ImageNet are shown in parentheses.

*ImageNet (IN) - classification:* approach for training on ImageNet with cross-entropy loss [18]. *Landmarks (SfM) - metric learning:* approach for metric learning with contrastive loss on image pairs obtained from Structure-from-Motion on landmarks [29]. *Artwork attributes (SemArt):* networks trained on the SemArt dataset [14] by Garcia *et al.* [13] for artwork attribute prediction. In particular, we consider variants for painting type (10 classes) or author (350 classes). *StylizedImageNet (SIN):* network trained by Geirhos *et al.* [15] on a stylized version of ImageNet to improve the texture bias of deep networks. *SwAV on ImageNet (IN) - self supervision:* representation learning on ImageNet with self-supervision by instance discrimination. The resulting network has achieved good results in concept generalization [6]. *Semi-weakly supervised (SWSL) on Instagram 1G + ImageNet:* teacher-student approach [41] with teacher pretrained on about 1 billion images with hashtags and student trained with teacher-generated pseudo-labels, eventually fine-tuned on ImageNet.

## 4   Experiments

We perform performance evaluation of the baseline approaches using GAP and accuracy on the test queries of the Met dataset. Training, if any, is performed on the training part of the Met, while the validation queries are either used as validation set during the training or to tune the hyper-parameters of the kNN classifier. Multi-scale representation and PCA whitening with dimensionality reduction to 512D are used unless otherwise stated.

**Image representation and kNN classifier components.** ResNet18 trained on ImageNet is used as backbone to perform recognition with a kNN classifier. Hyper-parameters $k$ and $\tau$ are tuned and reported separately per experiment in Table 4 which shows the impact of different components. The multi-scale representation and the use of whitening are essential parts of main approach (ID4 vs ID1,ID2, and ID3). Fixing $k = 1$ (ID5) is equivalent to no use of soft-max normalization and has significantly lower GAP on all queries, slightly lower GAP on Met queries, and identical accuracy by definition. Confidence normalization is therefore very important for handling distractors and high GAP performance. Finally, we show that having distractors in the validation set is boosting GAP by better kNN classifier hyper-parameter tuning (ID6 vs ID4).

| Method | GAP | GAP$^-$ | ACC |
|---|---|---|---|
| Parametric classification | | | |
| R18IN DNet CE | 9.6 | 24.7 | 30.6 |
| R18IN DNet AF | 16.9 | 32.0 | 36.6 |
| kNN classification | | | |
| R18IN (baseline) | 15.9 | 37.5 | 42.3 |
| R18IN DNet CE | 21.6 | 40.4 | 44.7 |
| R18IN DNet AF | 23.7 | 43.9 | 47.4 |
| R18IN SimSiam | 26.8 | 42.3 | 45.6 |
| R18IN Con-Syn | 30.4 | 46.6 | 49.4 |
| R18IN Con-Syn+Real | 29.8 | 46.0 | 48.8 |
| R18IN Con-Syn+Real-closest | **32.5** | **47.5** | **50.0** |
| R18SWSL (baseline) | 24.7 | 47.0 | 50.9 |
| R18SWSL Con-Syn+Real-closest | **36.1** | **52.4** | **55.0** |

Table 6: Performance comparison for different types of training on the Met dataset. Training starts from the result of pretraining on ImageNet or that of SWSL. Baseline: not trained on the Met.

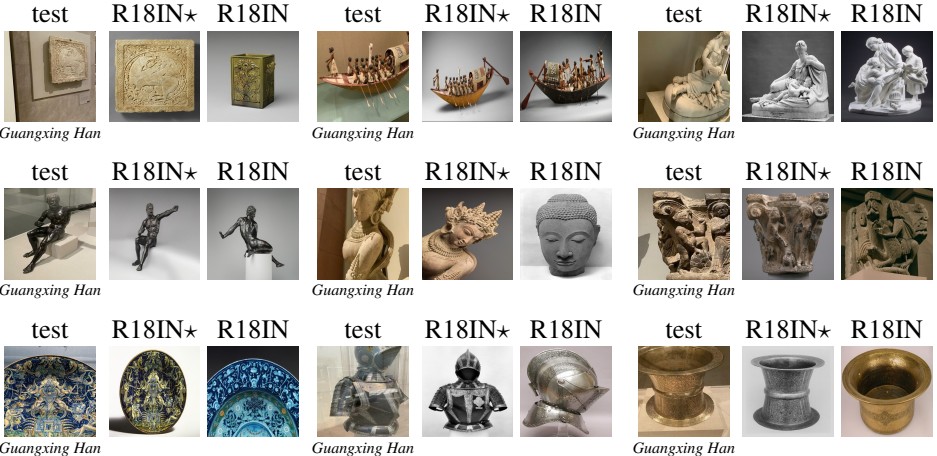

Figure 6: Examples of incorrect and correct classification of test images for R18IN (baseline) and R18IN Con-Syn+Real-closest (R18IN⋆), respectively. The test images are shown next to their nearest neighbor from the Met exhibits that produced the respective prediction per method.

**Pretrained backbones and kNN classifier.** Table 5 summarizes results of recognition performance with a kNN classifier for backbones pretrained on different tasks. Networks for art attribute prediction perform worse than the ImageNet ones, verifying that the task of art attribute prediction is far from that of ILR. The network for metric learning on landmarks provides improvements; despite the domain difference (artwork vs landmarks), training for metric learning well reflects the objectives of ILR. SwAV provides a performance boost, verifying the usefulness of unsupervised representation learning for better generalization. Finally, SWSL is the best performing variant demonstrating the benefits of learning on a very large image corpus despite the noisy labels; we expect the training set to include many artworks too.

**Training on the Met dataset.** Results from training on the Met dataset are shown in Table 6 with a parametric deep network classifier and with a kNN classifier. The latter is shown to be superior, while carrying the extra cost of storing a 512-D vector per training image. AF is shown to be better than CE, verifying prior results on ILR [4]. SimSiam improves the performance over the baseline without the use of any supervision indicating that self-supervised learning is a promising direction for ILR. Con-Syn uses the same positives as SimSiam but further boosts the performance by the use of negatives. Including real positives too with constrastive loss achieves the best performance but only if the positive pair is properly disambiguated (Real-closest vs Real). Improvements by training on the Met are confirmed starting from R18SWSL too. Examples where R18IN Con-Syn+Real-

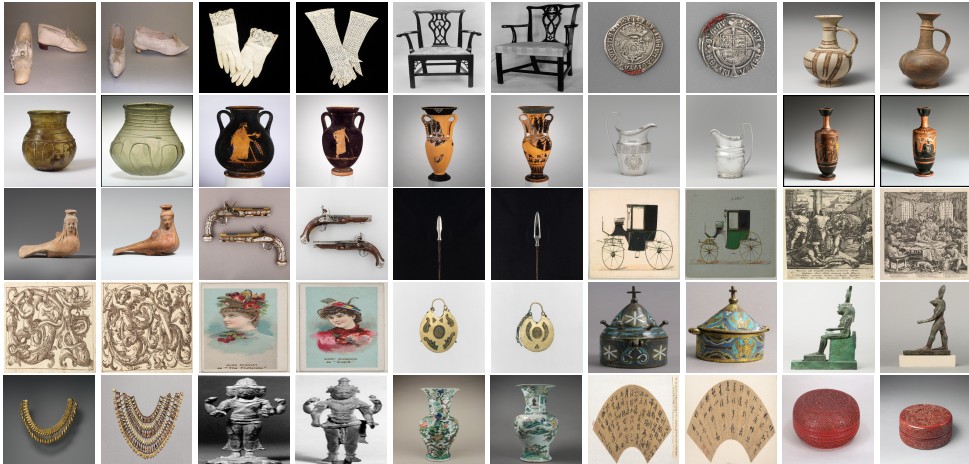

Figure 7: Examples of hard negative pairs formed by the approaches that use the Contrastive loss on the Met training set. These examples additionally demonstrate the large inter-class similarity of the dataset. Images are shown as squares only for the purposes of this figure.

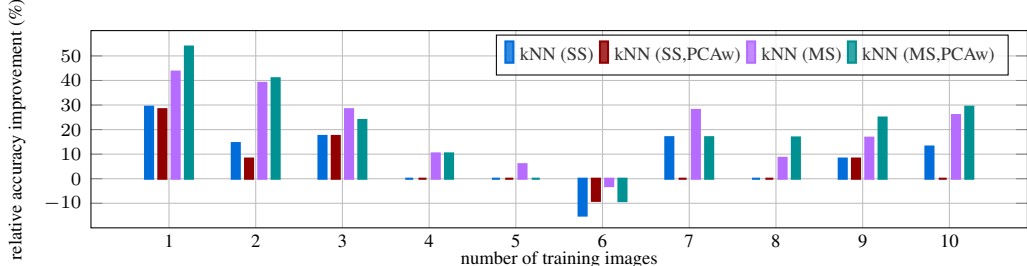

Figure 8: Accuracy improvement of the kNN classifier over the parametric one for varying number of training images per class. DNet is trained with AF loss for the parametric classifier, while the embeddings learned with this setup are used for the kNN classifier. Relative improvements are reported in percentage for the different embedding variants.

closest succeeds in prediction but the R18IN baseline fails are shown in Figure 6. These cases include challenges such as large view points changes and high inter-class similarity. Examples of hard negative pairs used in the contrastive variants are shown in Figure 7.

**Few training examples and kNN classifier.** We train a parametric classifier and additionally use the resulting embeddings for the kNN classifier. A comparison is shown in Figure 8, where performance is reported separately according to the number of training examples per ground-truth class of each query. The kNN classifier does not only perform better than the parametric one, but is shown to be more suitable for long tail recognition, as it achieves increasingly higher gains for more underrepresented classes.

## 5   Conclusions

This work introduces a new large-scale dataset for ILR on artworks. It is the first dataset on artworks to focus on this task, the only large-scale ILR dataset with clean annotations, and it poses a number of different challenges. The considered task is closer to ILR and deep representation learning than it is to popular computer vision tasks in the artwork domain, whilst including many of the same challenges. Fine-tuning the representation on The Met exhibits appears essential but also challenging due to the training set statistics. We expect this dataset to foster research not only on ILR for artworks but also for ILR across multiple domains, when combined with other existing datasets.

# 6 Acknowledgements

The authors would like to thank The Met employees Jennie Choi and Maria Kessler for their support and help, Andre Araujo, Tobias Weyand, and Xu Zhang for valuable discussions during the earlier stages of this work, and all the Flickr photographers whose photos are included in this dataset. This work was supported by JSPS KAKENHI No. JP20K19822, Junior Star GACR grant No. GM 21-28830M, and MSMT LL1901 ERC-CZ grant.

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
