# OpenReview forum: "The Met Dataset: Instance-level Recognition for Artworks"
_NeurIPS.cc/2021/Track/Datasets_and_Benchmarks/Round2 — NeurIPS 2021 Datasets and Benchmarks Track (Round 2)_

### Official Review · Reviewer_rsPe · 2021-09-20
**An Instance-level Recognition dataset that may not raise common interest from the audience of NeurIPS**

**Rating:** 5
**Confidence:** 4
**Clarity:** The paper is presented in a clear and…

**Strengths:**

1. The paper is large-scale in the domain of artworks and seldom datasets can be precisely labeled without noise at this scale.
2. Comparions to existing artwork datasets are well conducted, which provides clear position of the proposed dataset.

**Weaknesses:**

1. The dataset may not raise common interest to the potential audience of NeurIPS. Comparing to other instance-level classification tasks such as face verification, place recognition, the artwork recognition receives obviously less attention.
2.  The authors claim the dataset is long-tailed. But in evaluation, techniques used to address long-tailed distribution are not incorporated, which makes the evaluation incomplete.
3. The queries are collected from 39 visitors, which are relatively narrow and can make the evaluation biased.

**Additional Feedback:**

In terms of how to improve the paper, please refer to the above sections. More proper venues for this paper inclue ICMR, ACM Multimedia, and other vision related venues etc.

**Correctness:**

The dataset construction basically makes sense. My only concern is that the query images are relatively narrow as they are collected from only 39 visitors. In terms of experiments, more evaluations are expected in the flavour of long-tailed learning.

**Documentation:**

The dataset is well documented.

**Relation To Prior Work:**

Sufficient and clear discussion to prior work are provided.

**Summary And Contributions:**

This paper proposes a dataset for instance-level recognition in the domain of artworks. The dataset includes training data collected from MET organization under studio condictions, and query data composed of images from MET collected by users and distractors. The dataset is large-scale and precisely labeled with no noise. Comparisons to existing datasets are clearly illustrated, which provides a clear position of this dataset. Experiments are performed based on both kNN and cross-entropy classifiers and for kNN, state-of-the-art feature representation strategies are evaluated.

---

> ### Author Response · Authors · 2021-09-30
> **Response**
>
> We thank the reviewer for the helpful feedback. Our responses to the reviewer comments follow.
>
> > "1. The dataset may not raise common interest to the potential audience of NeurIPS. Comparing to other instance-level classification tasks such as face verification, place recognition, the artwork recognition receives obviously less attention."
>
> We believe that artwork ILR has not received attention because of the lack of relevant datasets. This is the gap that this paper is filling. As the reviewer points out "seldom datasets can be precisely labeled without noise at this scale", which is that we achieve in this work to be close to an interesting real world setup. The community has largely focused on urban scenery and buildings/landmarks (place recognition, GLDv2), while this is not the only domain that is relevant to ILR. Our dataset is aiming to follow the example of other tasks, such as fine-grained image classification (please see responses to reviewer yHmR).
>
>
> > "2. The authors claim the dataset is long-tailed. But in evaluation, techniques used to address long-tailed distribution are not incorporated, which makes the evaluation incomplete."
>
>
> The table below shows the average recognition accuracy vs the number of training images per class. It demonstrates that the non-parametric classifier suffers less from the imbalance than the parametric one; the relative improvement is larger in the low-shot regime.
>
> | #images/class  	 		| 	1 |   2 |   3 |   4 | 5   | 6   |   7 | 8   | 9   | 10  | avg |
> | ------------------------- |:---:|:---:|:---:|:---:|:---:|:---:|:---:|:---:|:---:|:---:|:---:|
> |A: DNET AF (prm cls)      | 31.4| 31.8| 36.8| 46.3| 32.1| 56.9| 48.6| 38.7| 52.2| 35.2| 36.6|
> |B: DNET AF (knn cls)      | 48.3| 44.8| 45.6| 51.1| 32.1| 51.7| 56.7| 45.2| 65.2| 45.4| 47.4|
> | relative improvement (%) of B vs A  | 53.8| 40.9| 23.9| 10.4|  0.0| -9.1| 16.7| 16.8| 24.9| 29.0|   29.5 |
>
> As part of the response we evaluated one of the variants in [Kang et al, DECOUPLING REPRESENTATION AND CLASSIFIER FOR LONG-TAILED RECOGNITION, ICLR 2021]. In particular, after the joint training, the representation is fixed, the classifier is re-initialized and trained with class balanced sampling. In our case this results in a 4% performance drop compared to the joint training. Additionally, joint training with class weighting (common approach for imbalanced datasets) does not offer any improvements. Note that the statistics of the MET dataset are different from the ones used in the work of Kang et al, which are commonly used datasets for long-tail recognition. In their datasets the ratio of the largest class size over smallest class size is up to 1,000 while it is equal to 10 in our case (see Fig 3 in the supplementary material for the underlined distribution). Finally, τ-normalization from the work of Kang et al is not necessary in our case, since we rely on a cosine similarity classifier where such a normalization is inherent. We will add all the above results and discussion in the final version of the paper.
>
>
> >"3. The queries are collected from 39 visitors, which are relatively narrow and can make the evaluation biased."
>
> We agree that the more visitors the better. However, note that all the Flickr users took photos with their own independent (artistic) style, without any instructions. This is not the case in datasets such as GLDv2 where external contractors were instructed to take photos of particular landmarks. Note that all photos needed to be under creative commons license, which significantly decreases the pool of candidate images.

---

> > ### Comment · Reviewer_rsPe · 2021-10-04
> > **agree to lift the score to boarderline accept**
> >
> > Dear Authors,
> >
> > Thank you for the detailed rebuttal. I echo my point again that I agree with the quality of the dataset but concern with the potential impact in the NeurIPS community. AC also holds similar view. But I have no objection in accepting the paper.

---

> > > ### Author Response · Authors · 2021-10-04
> > > **Response II**
> > >
> > > Thank you for the follow-up comment.
> > >
> > > We would appreciate some additional information regarding the reviewer’s thoughts on why they believe that the future impact will be small. Currently, artwork recognition receives less attention because of the lack of good datasets. Since reviewers agree about the good quality of our dataset, we expect the attention to increase with the introduction of our dataset.
> > >
> > > Both ILR and artworks have attracted attention from the NeurIPS community. In particular, artworks (and datasets thereof) have been commonly used within the NeurIPS community, as evidenced by the now 5th edition of the Creativity workshop [1], works on art and style [2, 3, 4], works that incorporate them for evaluation purposes [5, 6], and demo on artwork recognition [9]. Our dataset fits well with these lines of work, and within the larger trend of increasing attention to art within the AI community. Additionally, datasets for ILR are of interest to the community as illustrated by recent works using landmark datasets [7,8]. By following a similar setup as GLDv2 we aim to encourage authors to experiment with new ILR approaches on our dataset, even if they do not have a specific interest in art, to evaluate the domain specificity of their approach. Thereby increasing the potential impact of our work. Moreover, as of yet there is no existing dataset that bridges these two fields, our dataset fills this void, creating new opportunities for the ILR and art fields within the NeurIPS community, as well as opportunities for a new community at the intersection of these fields.
> > >
> > > [1] Machine Learning for Creativity and Design. https://neuripscreativityworkshop.github.io/2021/
> > > [2] Wynen et al. Unsupervised Learning of Artistic Styles with Archetypal Style Analysis. NeurIPS 2018.
> > > [3] Li et al. Universal Style Transfer via Feature Transforms. NeurIPS 2017.
> > > [4] Härkönen et al. GANSpace: Discovering Interpretable GAN Controls. NeurIPS 2020.
> > > [5] Mettes et al. Hyperspherical Prototype Networks. NeurIPS 2019.
> > > [6] Hung et al. Compacting, Picking and Growing for Unforgetting Continual Learning. NeurIPS 2019
> > > [7] Hoe et al. One Loss for All: Deep Hashing with a Single Cosine Similarity based Learning Objective. NeurIPS 2021
> > > [8] Liu et al. Guided Similarity Separation for Image Retrieval. NeurIPS 2019
> > > [9] Hamilton et al. MosAIc: Finding Artistic Connections across Culture with Conditional Image Retrieval, NeurIPS 2020

---

### Official Review · Reviewer_yHmR · 2021-09-20
**A well-constructed ILR dataset on artwork. More analysis on the difference between the proposed dataset and existing ILR dataset is required.**

**Rating:** 6
**Confidence:** 4
**Clarity:** This paper is very well written.

**Strengths:**

1. Various methods are compared in the benchmark. Results and training details are well provided.
2. The proposed dataset has a much larger number of images and classes than the existing artwork ILR dataset, making it reliable for further research works in the future.
3. This paper is well written with well-designed figures and tables.

**Weaknesses:**

The proposed dataset is very similar to existing ILR datasets, which may have difficulties in promoting new research works. GLDv2 has a similar number of instances but with a significantly bigger number of images. Although there are differences between the two datasets as described in the paper, how they will influence existing ILR methods is uncertain. More analysis on how specific features of the proposed dataset influence the existing ILR methods is required.

**Additional Feedback:**

None

**Correctness:**

The submission is a dataset with the benchmark which is constructed in a sound way compared with existing works. The benchmark covers different deep learning methods for proper evaluation.

**Documentation:**

The data collection and annotation process are described in detail.

**Ethics:**

No ethics issues.

**Relation To Prior Work:**

The difference between the proposed work and previous works is well discussed.

**Summary And Contributions:**

This well-written paper presents an ILR dataset on artworks and summarizes the comparison to the existing ILR dataset. Different kinds of methods are compared and evaluated properly in the paper.

---

> ### Author Response · Authors · 2021-09-30
> **Response**
>
> We thank the reviewer for the positive feedback. Our responses to the reviewer comments follow.
>
> >"The proposed dataset is very similar to existing ILR datasets, which may have difficulties in promoting new research works. GLDv2 has a similar number of instances but with a significantly bigger number of images. Although there are differences between the two datasets as described in the paper, how they will influence existing ILR methods is uncertain. More analysis on how specific features of the proposed dataset influence the existing ILR methods is required."
>
> We are relying on the largest available artwork collection that is appropriate for instance-level recognition. It corresponds to the collection of one of the biggest artwork museums world-widely which makes it a valuable test case.
>
> We would like to point out both the distinctive differences between the MET dataset and GLDv2, but also the usefulness of following a similar setup.
> 1. differences:
> - MET has accurate training labels, while GLDv2 includes noise due to crowdsourcing. Training on GLDv2 benefits from explicitly handling the noise aspect, while this is not required on the MET dataset. We expect this to allow future approaches to focus on representation learning and large-scale recognition approaches rather than training with noisy labels.
> - the percentage of classes that correspond to a single training image is roughly 5 times larger for MET than GLDv2. This makes self-supervised learning (SSL) approaches more relevant, as we show in the paper. SSL has lately attracted the attention of the community and the MET dataset indicates another relevant task that has not been considered in SSL so far.
> 2. similar setup:
> - Following a similar setup (dataset structure, dataset split, evaluation metrics) as GLDv2 is an advantage, which we count on to encourage the development of ILR approaches that are applicable beyond a specific domain. The same advantage has been demonstrated for other tasks too, such as fine-grained image classification with a variety of datasets (CUB birds, CARS 196, Stanford products, and more), where all these different datasets follow the same setup with each other and novel approaches are expected to evaluate and perform well in all of them.

---

> > ### Comment · Reviewer_yHmR · 2021-10-04
> > **Reply to authors**
> >
> > Thank you for the detailed response.
> > I agree with the authors that MET and GLDv2 are different in some aspects. However, the authors didn't perform enough experiments and compare the results between ones on MET and GLDv2 to show how these differences influence models' performance.
> >
> > The dataset is very well-constructed and I believe it can be useful for research on computer vision with artworks. However, it may not draw very much attention of the audience of NeurIPS.
> >
> > Thus, I will keep my positive review but not raising my score.

---

> > > ### Author Response · Authors · 2021-10-04
> > > **Response II**
> > >
> > > Thank you for the follow-up comment.
> > >
> > > > The dataset is very well-constructed and I believe it can be useful for research on computer vision with artworks. However, it may not draw very much attention of the audience of NeurIPS.
> > >
> > > Please see our latest response to Reviewer rsPe.
> > >
> > > >  I agree with the authors that MET and GLDv2 are different in some aspects. However, the authors didn't perform enough experiments and compare the results between ones on MET and GLDv2 to show how these differences influence models' performance.
> > >
> > > Please note that the MET dataset is not meant to replace GLDv2 but to complement it. We will target evaluating indicative methods from Table 6 on GLDv2 and include them in the camera ready version. We would appreciate suggestions for particular experiments to be included in the camera ready or our future work.

---

### Official Review · Reviewer_Q1T2 · 2021-09-21
**A dataset for artworks**

**Rating:** 7
**Confidence:** 3
**Correctness:** Looks correct.
**Clarity:** Yes.

**Strengths:**

1. The proposed dataset is a large-scale dataset consisting of about 400k images from more than 224k classes. The proposed dataset could be useful for testing algorithms for representation learning.
2. The authors also provide benchmarks and baseline models on the proposed dataset.


**Weaknesses:**

It would be great to provide more details on the labeling process. For instance, how long will it take to label the images?

**Additional Feedback:**

This work introduces a large-scale dataset for developing representation learning algorithms.

**Documentation:**

Yes.

**Ethics:**

No.

**Relation To Prior Work:**

Yes.

**Summary And Contributions:**

This work introduces a large-scale dataset for instance-level recognition for artworks. The authors provide benchmarks on the dataset.

---

> ### Author Response · Authors · 2021-09-30
> **Response**
>
> We thank the reviewer for the positive feedback. Our responses to the reviewer comments follow.
>
> > "It would be great to provide more details on the labeling process. For instance, how long will it take to label the images?"
>
> For each of the query annotation phases, we provide more details below:
> - The filtering phase took 4.92 seconds on average per candidate query and discarded about 52% of the images.
> - The annotation phase took 62.57 seconds on average per candidate query and discarded about 44% of the images.
> - The verification phase took 20.01 seconds on average per candidate query and per annotator (two annotators per query) and discarded about 55% of the remaining images.

---

### Official Review · Reviewer_VUdM · 2021-09-22
**Instance-level Recognition for Artworks: The MET Dataset**

**Rating:** 7
**Confidence:** 4
**Correctness:** Yes
**Clarity:** Yes

**Strengths:**

+ The MET dataset is the first large-scale dataset in the task of instance-level recognition of artworks.
+ The paper is written well. The collection and annotation of the dataset are described in detail.
+ The MET dataset poses a number of different challenges and has great significance in academic research.

**Weaknesses:**

- The MET dataset is collected from multiple sources. The authors are encouraged to evaluate the performance of domain adaptation.
-  It's confusing that the val/test set is very small both on image level and class level compared with the training set.

**Additional Feedback:**

See part "Weaknesses"

**Documentation:**

Yes

**Ethics:**

See part "Weaknesses"

**Relation To Prior Work:**

Yes

**Summary And Contributions:**

This paper proposes the MET dataset for instance-level recognition of artworks. The MET dataset poses a number of different challenges and has great significance in academic research.

---

> ### Author Response · Authors · 2021-09-30
> **Response**
>
> We thank the reviewer for the positive feedback. Our responses to the reviewer comments follow.
>
> > "The MET dataset is collected from multiple sources. The authors are encouraged to evaluate the performance of domain adaptation."
>
> Preliminary effort to use AdaBN [Li et al, Revisiting Batch Normalization For Practical Domain Adaptation] as a domain adaptation approach did not result in any improvements. The domains are artwork queries (val), non-artwork queries (val), and MET artworks (train). Note that the non-artwork set is highly diverse as it can include anything that is not an artwork.
>
> >"It's confusing that the val/test set is very small both on image level and class level compared with the training set."
>
> Including all 224k classes in the val and test set is a very hard task especially when considering accurate and manually verified ground truth. As a workaround, the MET dataset performs the performance evaluation only over a subset of the training classes. The task is still large-scale recognition with a very large category set during the training, but evaluation of recognition performance is approximated by evaluating only on a subset of the classes. The same choice is adopted in the widely used GLDv2.

---

### Decision · Program_Chairs · 2021-10-09

**Decision:**

Accept

**Comment:**

This paper presents the MET dataset, a cleanly-labeled large-scale image dataset in the artwork domain for the study of instance-level recognition in the domain of artworks. While reviewers agree that the dataset is well-constructed and the paper is clearly written, some raised the concerns that the scope of the dataset is somehow overlapping with GLDv2, and that the dataset seems to be of limited interest to the NeurIPS audiences because of it domain-specific nature. After the authors' response, one reviewer agreed to raise the score from 5 to 6 (borderline accept), although it is not reflected in the system. Then now the all four reviewers are in the positive side, two of them with clear accept.
I, the AC, agree that this paper will fit better to vision or multimedia conferences. Nonetheless, since there is no severe weakness in the paper, I think it is ready to be presented at this track.